# Transferring Pre-trained Multimodal Representations with Cross-modal Similarity Matching

**Byoungjip Kim**[1], **Sungik Choi**[1], **Dasol Hwang**[1], **Moontae Lee**[1,2], **Honglak Lee**[1]

LG AI Research[1], University of Illinois Chicago[2]

{bjkim, sungik.choi, dasol.hwang, moontae.lee, honglak}@lgresearch.ai

## Abstract

Despite surprising performance on zero-shot transfer, pre-training a large-scale multimodal model is often prohibitive as it requires a huge amount of data and computing resources. In this paper, we propose a method (**BeamCLIP**) that can effectively transfer the representations of a large pre-trained multimodal model (CLIP-ViT) into a small target model (e.g., ResNet-18). For unsupervised transfer, we introduce *cross-modal similarity matching* (CSM) that enables a student model to learn the representations of a teacher model by matching the relative similarity distribution across text prompt embeddings. To better encode the text prompts, we design *context-based prompt augmentation* (CPA) that can alleviate the lexical ambiguity of input text prompts. Our experiments show that unsupervised representation transfer of a pre-trained vision-language model enables a small ResNet-18 to achieve a better ImageNet-1K top-1 linear probe accuracy (66.2%) than vision-only self-supervised learning (SSL) methods (e.g., SimCLR: 51.8%, SwAV: 63.7%), while closing the gap with supervised learning (69.8%).

## 1 Introduction

Learning transferable representations is crucial for successful downstream tasks. Contrastive learning such as SimCLR [4] and MoCo-v2 [6] have shown notable success by forcing features of individual classes to be clustered and sufficiently scattered [41]. But their linear probe performances are still far behind the supervised learning as shown in Figure 1. Recently, large-scale vision and language pre-trained (VLP) models provide highly transferable visual representations via language supervision. However, learning VLP models from scratch is prohibitive as it requires large amounts of training data and computing resources. For example, training CLIP [32] requires 400M paired image-text data and several hundreds of GPUs. ALIGN [22] further scales up to leverage alternative texts specified for descriptions of web images. While these models are often based on large Transformers [40], small ConvNets such as ResNet-50 [17]

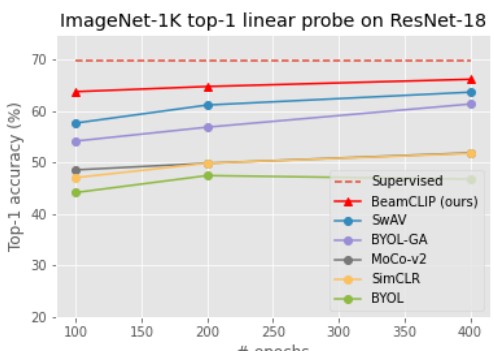

Figure 1: **ImageNet-1K top-1 linear probe accuracy on ResNet-18 representations.** By transferring CLIP-ViT [32] vision-language representations to ResNet-18, the BeamCLIP can learn better visual representations than vision-only self-supervised learning (SSL) methods in terms of the linear probe accuracy.

and MobileNet [20] are still widely used in practice [1] and even more crucial for low-resource

36th Conference on Neural Information Processing Systems (NeurIPS 2022).

environments. We reformulate representation learning in terms of knowledge transfer from a large pre-trained model to a small practical model.

Large-scale vision-language pre-trained models exhibit strong alignments between different modalities. CLIP [32] learns visual concepts from natural language supervision, mapping image and text into the same vector space. As their training data is not only huge but inaccessible, however, conventional knowledge distillation [19] based on the source training data is no longer a viable option. Instead, we propose *cross-modal similarity matching* (CSM). Imagine your goal is to learn a high quality representation for an input dog image in CIFAR-10 [23] as in Figure 2. Since CLIP was trained on numerous image-caption pairs, angular distances from the dog image embedding to the caption embeddings of other anchor prompt texts such as `"A photo of cat"` or `"A photo of horse"` must comprehensively preserve their visual differences. By training the student to preserve angular relations witnessed from the teacher, our model achieves near benchmark performance without accessing the original data for training CLIP.

To better encode the text prompts, we design *context-based prompt augmentation* (CPA) that can alleviate the lexical ambiguity of the input text prompts. We find that lexical ambiguity in prompt texts can lead to semantically incorrect text embeddings. This may result in unexpected discrepancies of image-text alignment in the teacher's embedding space. Also, it is known that the zero-shot performance of CLIP can be improved by designing task-specific prompt texts. Inspired by this, we design CPA that extends the basic prompt of CLIP to better encode prototypical anchor representations.

Our experimental results show that the **BeamCLIP** ("beam" means to transmit) achieves the strongest and near benchmark performance on ImageNet-1K [10] top-1 linear probe accuracy when using most popular ResNet-18 and ResNet-50 as the student network. We also compare the effectiveness of the BeamCLIP against zero-shot transfer learning. Further, we provide ablation study results to show how much each component contributes to the performance.

Contributions of this paper can be summarized as follows:

- We propose a method (**BeamCLIP**) that can effectively transfer the representations of a large pre-trained multimodal model (e.g., CLIP-ViT) into a small target model (e.g., ResNet-18 or ResNet-50). To achieve this, we introduce *cross-modal similarity matching* (CSM) and *context-based prompt augmentation* (CPA). (Figure 2).

- We empirically show that BeamCLIP enables a small target model (e.g., ResNet-18) to achieve a better ImageNet-1K linear probe accuracy than vision-only self-supervised learning (SSL) methods, by effectively transferring CLIP-ViT representations. (Figure 1, Table 2, and Table 3.

- We also explore the zero-shot capability of the BeamCLIP (Table 5) and analyze the effectiveness of the BeamCLIP on various target datasets (Table 6 and Table 7).

## 2 Related Work

**Vision and language pre-trainig.** Vision and language pre-training (VLP) aims to jointly learn vision and language representations that can be transferred to the downstream tasks such as visual question answering (VQA), image captioning, and vision and language navigation (VLN). There are BERT-based vision and language models such as VLBERT [35], ViLBERT [26], and UNITER [7]. Also, there are contrastive learning-based models such as CLIP [32] and ALIGN [22]. These models use contrastive loss [30] to learn aligned vision and language representations by performing a task of matching a large-scale image and text pairs. The BeamCLIP aims to transfer the rich representations of large-scale vision and language pre-trained models such as CLIP and ALIGN to a small target model.

**Self-supervised learning.** Self-supervised learning (SSL) aims to learn highly transferable representations by using unlabeled data. In computer vision, at the early stage, task-specific self-supervised methods were introduced. These include Context Prediction [11], Rotation Prediction [14], and Colorization [43]. More recently, contrastive learning-based methods were introduced as a task-agnostic approach. These include SimCLR [4] and MoCo-v2 [6]. However, since contrastive self-supervised methods require a large batch size, non-contrastive methods have been introduced. These include SwAV [3], BYOL [16], and SimSiam [5]. In this paper, we empirically show that the BeamCLIP can

provide better visual representations than the state-of-the-art SSL methods by leveraging a large-scale pre-trained multimodal model.

**Knowledge distillation.**   Knowledge distillation (KD) [19] aims to transfer rich knowledge from a strong teacher model to a target student model. In a conventional setting, it encourages the student model to mimic the task-specific prediction of the teacher model. As the student model is trained to predict the same probability distribution over pre-defined classes as the teacher model's, using Kullback-Leibler (KL) divergence is a natural metric to measure the error between the two models. For a classification task, the loss function can be formulated as follows:

$$\mathcal{L}_{\text{KD}} = \sum_i H(p_i, q_i^S) + \sum_i KL(p_i^T \| p_i^S). \tag{1}$$

The first term indicates the supervised loss, where $p_i$ denotes the one-hot labels and $H(p, q)$ denotes cross-entropy. The second term is the distillation loss, where $p_i^T$ and $p_i^S$ are the softmax predictions of the teacher and student models, respectively.

**Similarity-based knowledge distillation.**   Recently, similarity-based knowledge distillation such as SEED [13], OSS [8], and ISD [37] was introduced in the context of self-supervised learning (SSL). SEED [13] showed that the linear probe accuracy of a small student (ResNet-18) can be improved by transferring the representations of a larger teacher (ResNet-50) pre-trained by SSL methods such as MoCo-v2 [6]. Unlike this, OSS [8] aims to transfer representations of an evolving teacher (ResNet-50) into a smaller student (ResNet-18) on the fly. Unlike SEED and OSS, ISD [37] considered the same size student and teacher network (ResNet-18), and showed a student can learn visual representations by iterativly distilling the similarity of teacher's representations. These works are closely related to our work. Unlike these works, the BeamCLIP aims to transfer rich vision and language representations of large-scale pre-trained models such as CLIP-ViT/16 [32] into a smaller network such as ResNet-18.

**Prompt engineering.**   Recently, researchers showed that prompt engineering [2] is surprisingly effective at improving the performance of large-scale language models (LLMs) on downstream tasks without fine-tuning. Prompts are input texts of language models that usually consist of a task description or several examples. To further simplify prompt engineering, prompt tuning [24] proposed to add $k$ learnable tokens to the input texts, while having language models frozen. Similar to GPT-3, it is known that the zero-shot performance of CLIP [32] can be improved by designing the prompt texts to each task. For example, on satellite image classification datasets, `"A satellite photo of a {label}"` provides better performance than the default `"A photo of a {label}"`. Inspired by this, we propose context-based prompt augmentation that extends the basic prompt of CLIP to better encode prototypical text anchor representations by alleviating the lexical ambiguity of class label texts.

## 3   Method

**Problem formulation.**   Formally, our problem is to transfer aligned cross-modal representations of a strong teacher model $f_\phi^T(\cdot)$ into a target student model $f_\theta^S(\cdot)$ with unlabeled data $\mathcal{D}_u = \{x_i\}_{i=1}^N$. Given each unlabeled image $x_i$, we formulate representation transfer as a regression task that matches teacher representations $f_\phi^T(x_i)$ to a student's $f_\theta^S(x_i)$. As the student network is parameterized by $\theta$, the learning objective is

$$\arg\min_\theta \sum_i^N \|f_\theta^S(x_i) - f_\phi^T(x_i)\|_2^2. \tag{2}$$

Normalizing the representations via $l_2$-normalization (*i.e.*, $q_i = f_\theta^S(x_i)/\|f_\theta^S(x_i)\|_2$ and $k_i = f_\phi^T(x_i)/\|f_\phi^T(x_i)\|_2$) leads to the following simplification:

$$\arg\min_\theta \sum_i^N \|q_i - k_i\|_2^2 = \arg\min_\theta \sum_i^N (2 - 2q_i \cdot k_i). \tag{3}$$

The problem now involves maximizing the cosine similarity between $l_2$-normalized representations from teacher and student models.

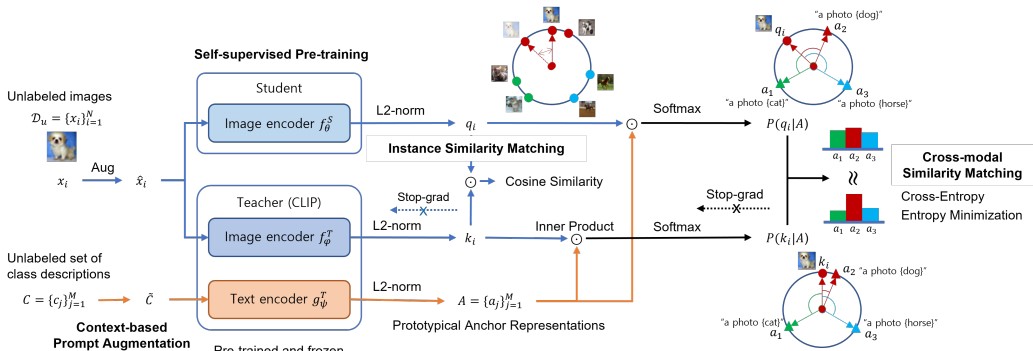

Figure 2: **Overview of the BeamCLIP.** Representation transfer can be viewed as a task in which, given a query input, a student model learns to regress a vector representation of a teacher model. The BeamCLIP first measures the normalized cross-modal similarity of the query image compared to anchor text representations in the teacher's embedding space. Then, it encourages the student to mimic the same cross-modal similarity in the student's embedding space. To better align image representations, our method uses self-supervised pre-training of the student model. Finally, to avoid text ambiguity, we uses context-based prompt augmentation.

**Method overview.** The overview of **BeamCLIP** is shown in Figure 2. The teacher model of the BeamCLIP consists of an image encoder $f_\phi^T(\cdot)$ and a text encoder $g_\psi^T(\cdot)$. These encoders are pre-trained under a simple task of matching images to texts with large-scale corpora. Image representations $f_\phi^T(x_i)$ and text representations $g_\psi^T(t_i)$ are thus well-aligned within a cross-modal embedding space. We provide the details of the BeamCLIP in the following sections. More specifically, we describe how to extend the basic problem setting by leveraging the unique features of CLIP where vision and language representations are precisely aligned. Throughout the paper, we use the notation CLIP-ViT/16 to denote the CLIP [32] model that uses Vision Transformer (ViT) [12] with the patch size of 16x16 as the image encoder. Similar to this, CLIP-RN50 denotes the CLIP model with ResNet-50 [17] as the image encoder.

### 3.1 Similarity-based cross-modal representation transfer

To effectively distill cross-modal representations, we use similarity-based matching as described above. Our similarity-based representation transfer utilizes two carefully designed loss functions: (1) instance similarity matching (ISM) loss and (2) cross-modal similarity matching (CSM) loss.

**Instance similarity matching.** This objective is directly derived from Eq. 3. Given a query image $x_i$, it encourages the student image encoder $f_\theta^S(\cdot)$ to regress the representation of the teacher image encoder $f_\phi^T(\cdot)$. We apply conventional image augmentations (see Appendix B.1) on a query image $x_i$, and the same augmented image $\hat{x}_i$ is fed to both the teacher and student image encoders. Given unlabeled query images $\mathcal{D}_u = \{x_i\}_{i=1}^N$, it is formulated as follows:

$$\mathcal{L}_{\text{ISM}} = -\sum_{i=1}^N \left( \frac{f_\theta^S(\hat{x}_i)}{\|f_\theta^S(\hat{x}_i)\|_2} \cdot \frac{f_\phi^T(\hat{x}_i)}{\|f_\phi^T(\hat{x}_i)\|_2} \right) = -\sum_{i=1}^N (q_i \cdot k_i). \tag{4}$$

However, the similarity signal from a single instance is not enough to constrain the student representations. For example, the topological ambiguity may occur in image encoding, since two symmetric representations have the same cosine similarity compared to a single teacher representation (see Appendix B.2). We conjecture that this can be mitigated by incorporating multiple anchor points to the query points. Based on this idea, we introduce cross-modal similarity matching loss.

**Cross-modal similarity matching.** To better align a student representation $q_i$ with the teacher representation $k_i$, we introduce cross-modal similarity matching (CSM) loss. We use multiple anchor points to cope with the ambiguity problem mentioned above. Further, we use text representations as anchor points, since we can easily generate prototypical anchor points by using text prompts and class

Table 1: Examples of context-based prompt augmentation for ambiguous class labels on Flowers102.

| Label Index | Label Name | Text Prompt |
|:-----------:|------------|-------------|
| 7 | bird of paradise | `"A photo of {bird of paradise}. {Strelitzia is a genus of five species of perennial plants, native to South Africa. It belongs to the plant family Strelitziaceae}."` |
| 10 | snapdragon | `"A photo of {snapdragon}. {Antirrhinum is a genus of plants commonly known as dragon flowers or snapdragons because of the flowers' fancied resemblance to the face of a dragon that opens and closes its mouth when laterally squeezed}."` |

labels. Since image and text representations are precisely aligned in CLIP, we can effectively apply this approach. More specifically, the BeamCLIP first measures the normalized image-text similarity of the query image compared to prototypical text points in the teacher's embedding space. Then, it encourages the student to mimic the same image-text similarity in the student's embedding space.

More formally, we generate multiple anchor representations $A = \{a_j\}_{j=1}^{M}$ by encoding class texts $C = \{c_j\}_{j=1}^{M}$ with the teacher text encoder $g_\psi^T(\cdot)$ (in other words, $a_j = g_\psi^T(c_j)$). To measure the similarity regarding to multiple anchor representations $A$, we define the normalized cross-modal similarity as follows:

$$s_j(k_i, A) = \frac{\exp\left((k_i \cdot a_j)/\tau\right)}{\sum_{m=1}^{M} \exp\left((k_i \cdot a_m)/\tau\right)} \qquad (5)$$

where $\tau$ is a temperature hyperparameter that is set to 0.01 in our experiments.

Then, we evaluate the cross-modal similarity distribution by using a set of normalized cross-modal similarities:

$$P(k_i|A) = [s_1(k_i, A), ..., s_M(k_i, A)]. \qquad (6)$$

Then, the student model is optimized to mimic the normalized cross-modal similarity of the teacher's embedding space by minimizing the cross entropy, *i.e.,* $H(P(k_i|A), P(q_i|A))$.

We further minimize the entropy of normalized cross-modal similarities in the student embedding space *i.e.,* $H(P(q_i|A))$. This minimization helps the student provide query representations $q_i$ that are more attracted to anchor representations $A = \{a_j\}_{j=1}^{M}$. This entropy minimization is also known to be effective in other domains such as semi-supervised learning [15, 29].

Altogether, the CSM loss is formulated as follows:

$$\mathcal{L}_{\text{CSM}} = \sum_{i=1}^{N} H(P(k_i|A), P(q_i|A)) + \sum_{i=1}^{N} H(P(q_i|A)). \qquad (7)$$

**Final Loss.** The final loss of the BeamCLIP is formulated as follows:

$$\mathcal{L}_{\text{BeamCLIP}} = \mathcal{L}_{\text{CSM}} + \lambda_{\text{ISM}}\mathcal{L}_{\text{ISM}} \qquad (8)$$

where $\lambda_{\text{ISM}}$ is the scale hyperparameter that is set to 10 in our experiments.

## 3.2 Context-based prompt augmentation

We found that lexical ambiguity in prompt texts can lead to semantically incorrect text embeddings. This may result in an unexpected discrepancy of image-text alignment in the teacher's embedding space. For example, Flowers102 [28] dataset has some classes with unusual and ambiguous flower names, such as "snapdragon", "bird of paradise", and "colt's foot".[1] Therefore, incorrect prototypical anchor points might be compared with a query image. To address this issue of semantic ambiguities in the text, we introduce context-based prompt augmentation (CPA), a data-driven approach that augments basic prompts with contextual text such as Wikipedia descriptions or hierarchical labels.

---

[1]Examples can be found at: 102 Category Flower Dataset.

For prompt tuning with Wikipedia descriptions, we use the template "A photo of a {label}. {Wikipedia description}". We use this template for Flowers102 and Pets37 in our experiments. We provide some examples from the Flowers102 dataset in Table 1. For prompt tuning with hierarchical labels, we use the template "A photo of a {fine label}, categorized as {coarse label}". We use this template for CIFAR100 and ImageNet in our experiments. Analogous examples from CIFAR100 can be found in Table 11 in Appendix B.3.

### 3.3 Other details

**Self-supervised pre-training of student.** To help the student mimic the teacher's cross-modal embedding space better, we pre-train the student image encoder with a self-supervised method. Since self-supervised pre-training such as SimCLR [4], MoCo-v2 [6], and SwAV [3] provides a weakly clustered embedding space based on similarities, it can be used as a better initial state for the student to mimic the teacher's embedding space. The details can be found in B.4. We show the effect of SSL pre-training of the student in the experiment section (see Table 4 and 7).

**Optimization.** For optimization we use SGD with cosine annealing schedule (SGDR) [25]. To stabilize training, we use a momentum encoder that updates its weights via exponential moving average (EMA) [18, 16]. The momentum encoder of a student $\theta_{\hat{S}}$ is updated using the following rule:

$$\theta_{\hat{S}} \leftarrow m\theta_{\hat{S}} + (1-m)\theta_S \tag{9}$$

where $\theta_S$ is the image encoder of a student model and $m$ is a momentum hyperparameter that is set 0.99 in our experiments. The model hyperparameters are summarized in Table 12 in Appendix B.5.

## 4 Experiments

**Downstream datasets.** We evaluate the BeamCLIP on six standard benchmark datasets: CIFAR10 [23], CIFAR100 [23], STL10 [9], Flowers102 [28], Pets37 [31], and ImageNet-1K [10]. Following convention, we split the datasets into train, validation, and test sets. Then, we use train set for transfer, and test set for evaluation. For ImageNet, we use the validation set as a test set, since its test set does not provide labels. More details on the datasets are summarized in Table 8 in Appendix A.

### 4.1 Representation transfer with unlabeled target data

**Setting.** We compare the BeamCLIP with various self-supervised methods in terms of linear probe accuracy on ImageNet-1K. Following the conventional protocol, we use ResNet-18 and ResNet-50 [17] as the base encoder and evaluate the learned representations by using logistic regression. We use LBFGS algorithm [44] for logistic regression. Its hyperparameter $C$ is determined through coarse-grained hyperparameter search on the validation split. And, the accuracy is evaluated in the test split. We found that it provides the best linear probe accuracy when $C$ is set to 30. We perform our experiments on 8 NVIDA A100 GPUs and it takes about 30 hours for 200 epoch training.

Table 2: **ImageNet-1K top-1 linear probe accuracy on ResNet-50**. We compare the BeamCLIP with vision-only self-supervised methods in terms of linear probe accuracy on ImageNet-1K. The BeamCLIP representations provide higher linear probe accuracy than self-supervised methods. This means better transferability. The values are quoted from the original paper, and n/a means "not available" from the paper.

| Method | Teacher | Student | Batch | Epochs | | |
|---|---|---|---|---|---|---|
| | | | | 200 | 400 | 800 |
| Supervised | ✗ | RN50 | 256 | | 76.2 | |
| SimCLR [4] | ✗ | RN50 | 512 | 65.6 | 66.7 | 67.4 |
| MoCo-v2 [6] | ✗ | RN50 | 256 | 67.5 | 70.1 | 71.1 |
| BYOL-GA [16] | ✗ | RN50 | 4096 | 70.6 | n/a | n/a |
| SwAV [3] | ✗ | RN50 | 256 | 72.0 | 74.3 | n/a |
| **BeamCLIP** (ours) | CLIP ViT-B/16 | RN50 | 512 | **74.8** | **75.1** | **75.0** |

Table 3: **ImageNet-1K top-1 linear probe accuracy on ResNet-18**.

| Method | Teacher | Student | Batch | Epochs | | |
|---|---|---|---|---|---|---|
| | | | | 100 | 200 | 400 |
| Supervised | ✗ | RN18 | 256 | | 69.8 | |
| SimCLR [4] | ✗ | RN18 | 256 | 47.1 | 49.9 | 51.8 |
| MoCo-V2 [6] | ✗ | RN18 | 256 | 48.6 | 49.9 | 51.9 |
| BYOL [16] | ✗ | RN18 | 256 | 44.2 | 47.5 | 46.8 |
| BYOL-GA [16] | ✗ | RN18 | 256 | 54.2 | 56.9 | 61.4 |
| SwAV [3] | ✗ | RN18 | 256 | 57.7 | 61.2 | 63.7 |
| OSS [8] | SSL RN50 | RN18 | 256 | 60.0 | 64.1 | 65.8 |
| **BeamCLIP** (ours) | CLIP ViT-B/16 | RN18 | 256 | **63.8** | **64.8** | **66.2** |

Table 4: **Effect of self-supervised pre-training of student.**

| Method | Teacher | Student | Pre-training of Student | Batch | Epoch | Linear Probe |
|---|---|---|---|---|---|---|
| Supervised | ✗ | RN50 | ✗ | 256 | - | 76.2 |
| **BeamCLIP** (ours) | CLIP ViT-B/16 | RN50 | SimCLR [4] | 512 | 200 | 74.8 |
| **BeamCLIP** (ours) | CLIP ViT-B/16 | RN50 | SwAV [3] | 512 | 200 | **75.8** |

**Transfer to ResNet-50.** Table 2 shows the comparison of the BeamCLIP with vision-only self-supervised methods such as SimCLR [4], MoCo-v2 [6], SwAV [3], BYOL [16], and SimSiam [5]. The BeamCLIP provides better visual representation by achieving 74.8% top-1 linear probe accuracy on ImageNet-1K [10]. While self-supervised methods take long training epochs to achieve comparable accuracy, BeamCLIP-RN50 achieves better accuracy with less training epochs. Also, note that BeamCLIP-RN50's representations provide better accuracy than CLIP-RN50's representations.

**Transfer to ResNet-18.** To check if the BeamCLIP can transfer CLIP representations into smaller models than ResNet-50 (24M), we also measure ImageNet-1K top-1 linear probe accuracy on ResNet-18 (11M). ResNet-18 is trained from scratch (not self-supervised pre-trained with SimCLR), while transferring CLIP ViT-B/16 representations. As shown in Table 3, BeamCLIP learns better representations than SSL methods such as SimCLR [4], MoCo-v2 [6], BYOL [16], and SwAV [3]. More importantly, the BeamCLIP provides better performance than OSS [8] that simultaneously learns and transfers representations from ResNet-50. The learning curve is presented in Figure 8 in Appendix C.1.

**Effect of self-supervised pre-training.** Table 4 shows ImageNet-1K top-1 linear probe accuracy on BeamCLIP-RN50 representations by using different SSL pre-training. With the better SSL method (SwAV [3] > SimCLR [4]), the BeamCLIP can learn better representations with an increased linear probe accuracy.

## 4.2 Representation transfer with unlabeled non-target data

To check if the BeamCLIP also inherits the powerful zero-shot capability of CLIP, we compare zero-shot accuracy of CLIP variants on ImageNet-1K. For zero-shot measure, we use CC-3M [34] and ImageNet-21K (12M samples) [33] that do not have overlap with ImageNet-1K. Table 5 shows the comparison of zero-shot accuracy. The BeamCLIP-RN50 achieves about 57.5% zero-shot accuracy that is highly comparable with CLIP RN-50 (59.6%). The learning curve is presented in Figure 9 of Appendix C.1.

## 4.3 Transfer learning accuracy on various target datasets

**Setting.** We evaluate how effectively the BeamCLIP transfers CLIP-ViT representations into a student model by evaluating classification accuracy on various datasets (see Table 6). We choose ResNet-50 [17] as the student model to compare the distilled target model with CLIP-RN50. Also, ResNet-50 is conventionally used in evaluating linear probe accuracy on ImageNet-1K. However, the BeamCLIP can adopt any architecture, not just ResNet-50.

Table 5: **Comparison of zero-shot accuracy on ImageNet-1K.** On ImageNet-1K, the BeamCLIP RN50 achieves about 57.5% zero-shot accuracy that is higly comparable with CLIP RN-50 (59.6%). To achieve such a high zero-shot accuracy, CLIP uses very large image-text pair data (WIT-400M). Instead, the BeamCLIP can achieve the comparable zero-shot accuracy by effectively transferring the teacher's representations, while using only 3% data (ImageNet-21K (12M)). Note that OpenCLIP provides only about 36.5% zero-shot accuracy with the similar amount of data (CC-12M).

| Method | Image Encoder | Training Data | Teacher Model | Text Prompts | Batch Size | Epochs | ImageNet Zero-shot |
|---|---|---|---|---|---|---|---|
| CLIP [32] | ViT-B/16 | WIT-400M | ✗ | ✗ | 32,768 | 32 | 68.6 |
| CLIP [32] | RN50 | WIT-400M | ✗ | ✗ | 32,768 | 32 | **59.6** |
| OpenCLIP [21] | RN50 | YFCC-15M | ✗ | ✗ | 256 ∗ 8 | 32 | 32.7 |
| OpenCLIP [21] | RN50 | CC-12M | ✗ | ✗ | 256 ∗ 8 | 32 | 36.5 |
| **BeamCLIP** (ours) | RN50 | CC-3M | CLIP ViT-B/16 | IN-1K | 64 ∗ 8 | 100 | 49.5 |
| **BeamCLIP** (ours) | RN50 | IN-21K (12M) | CLIP ViT-B/16 | IN-1K | 64 ∗ 8 | 50 | 53.6 |
| **BeamCLIP** (ours) | RN50 | IN-21K (12M) | CLIP ViT-B/16 | IN-1K | 64 ∗ 8 | 200 | 57.5 |

As baselines, we choose two representative distillation methods among many methods: (1) conventional knowledge distillation (KD) [19] and (2) contrastive representation distillation (CRD) [38]. Since conventional KD aims to mimic the task-specific predictions of the teacher model unlike the BeamCLIP , we apply the KL divergence on minimizing the cross-modal similarity distribution (*i.e.,* $P(q_i|A)$ and $P(k_i|A)$), instead of the Cross-entropy (CE). CRD proposes a variant of InfoNCE loss for representation distillation, which we apply on normalized representations (*i.e., $q_i$ and $k_i$*). The details of each method can be found in the related work section 2.

**Results.** Table 6 shows a comparison of teacher and student accuracy on various datasets. We empirically demonstrate that the BeamCLIP can effectively transfer vision and language representations of a large teacher model (CLIP ViT-B/16) into a small student model (ResNet-50). We find that the KL divergence used in conventional knowledge distillation (KD) is not effective in transferring CLIP-ViT representations. Also, the contrastive learning-based approach is not effective. Unlike this, the BeamCLIP can effectively transfer CLIP ViT-B/16 representations into ResNet-50, achieving very high accuracy that is comparable or better than the teacher accuracy. KD simply minimizes the error between single instances. We conjecture that the cross-modal similarity to multiple anchor points introduced in the BeamCLIP helps the student preserve the topology of the teacher's embedding space.

Also, note that context-based prompt augmentation helps achieve better accuracy after representation transfer. Since Flowers102 has many ambiguous labels, our experiment shows that text prompt augmentation significantly increases the student's accuracy compared to the teacher's accuracy.

**Ablation study.** Table 7 shows the ablation study results of the BeamCLIP . Our empirical findings are as follows: (1) Instance similarity matching (ISM) is not enough by itself to preserve the topology

Table 6: **Comparison of teacher and student accuracy on various datasets.** Conventional knowledge distillation with the KL divergence is not effective in transferring CLIP-ViT representations. In contrast, the BeamCLIP effectively transfers CLIP ViT-B/16 representations into ResNet-50 by using unlabeled query data. We denoted with ∗ in cases our student model surpasses the accuracy of the teacher model.

| Method | Model Type | Img. Enc. | Param. Size | CIFAR10 | CIFAR100 | STL10 | Flowers102 | Pets37 | ImageNet-1K |
|---|---|---|---|---|---|---|---|---|---|
| CLIP [32] (zero-shot) | T | ViT-B/16 | 76M | 91.6 | **68.7** | **98.2** | 70.4 | **88.9** | **68.6** |
| CRD [38] | S | RN50 | 24M | 76.78 | 38.83 | 81.41 | 26.15 | 62.22 | 30.90 |
| KD [19] | S | RN50 | 24M | 90.89 | 58.02 | 93.28 | 49.01 | 77.51 | 56.38 |
| **BeamCLIP** (ours) | S | RN50 | 24M | **92.10**∗ | 67.35 | 97.45 | **75.86**∗ | 86.94 | 66.17 |

Table 7: **Ablation study results.** The acronym denotes the sub-methods introduces in the BeamCLIP : (1-1) ISM means instance similarity matching loss, (1-2) CSM means cross-modal similarity matching loss, (2) CPA means context-based prompt augmentation, and (3) SSL PT means self-supervised pre-training of student, . All the technical components contribute to the improvements of transfer learning accuracy at the student. Note that we only perform CPA on datasets with more than 100 of class labels.

| Method | Type | Img. Enc. | (1-1) ISM | (1-2) CSM | (2) CPA | (3) SSL PT | CIFAR10 | CIFAR100 | Flowers102 | ImageNet-1K |
|---|---|---|---|---|---|---|---|---|---|---|
| CLIP [32] (zero-shot) | T | ViT-B/16 | - | - | - | - | 91.6 | **68.7** | 70.4 | **68.6** |
| Unsupervised Representation Transfer (**BeamCLIP**) | S | RN50 | Cosine | ✗ | ✗ | ✗ | 39.07 | 6.04 | 1.43 | 21.83 |
| | S | RN50 | Cosine | ✗ | ✗ | SimCLR | 87.00 | 48.26 | 3.59 | 51.83 |
| | S | RN50 | ✗ | CE | ✗ | ✗ | 91.28 | 58.90 | 12.94 | 63.30 |
| | S | RN50 | ✗ | CE | ✗ | SimCLR | 91.53 | 65.64 | 62.18 | 65.45 |
| | S | RN50 | ✗ | CE+EntMin | ✗ | SimCLR | 91.71 | 66.14 | 63.96 | **66.23** |
| | S | RN50 | Cosine | CE+EntMin | ✗ | SimCLR | **92.10**$^*$ | 66.18 | 64.14 | 65.76 |
| | S | RN50 | Cosine | CE+EntMin | ✓ | SimCLR | - | 67.35 | **75.86**$^*$ | 66.17 |

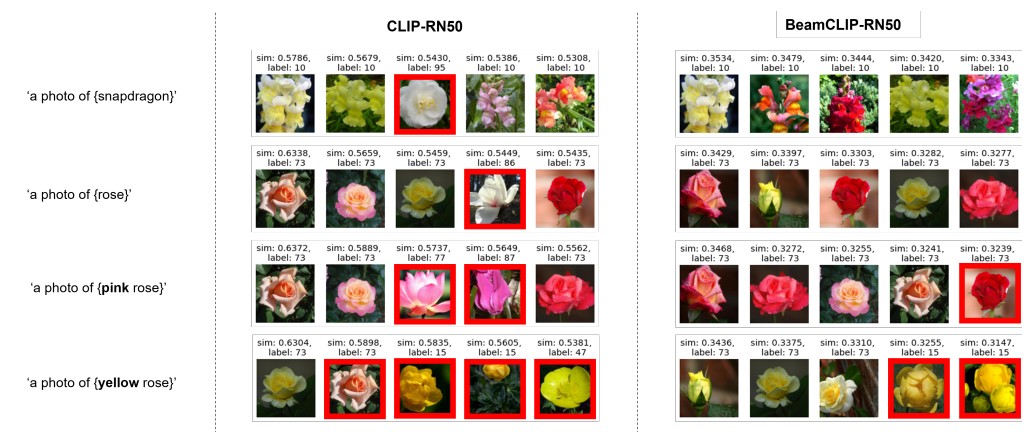

Figure 3: **Comparison of CLIP-RN50 and BeamCLIP-RN50.** This figure shows the top-5 text-image retrieval results. A red rectangle denotes an incorrect result. CLIP-RN50 provides many incorrect results, since its zero-shot accuracy is relatively low. In contrast, BeamCLIP-RN50 provides much improved results, since it is transferred from CLIP-ViT/16 with higher zero-shot accuracy.

of the teacher's embedding space. (2) Cross-modal similarity matching (SCM) compared to multiple anchor points helps the student mimic the teacher's embedding space. (3) Self-supervised pre-training of the student (SSL PT) helps the student mimic the teacher's embedding space. (4) Entropy minimization (EntMin) helps to improve the accuracy. (5) Context-based prompt augmentation (CPA) helps measure the similarity more precisely. As shown in the table, Flowers102 dataset is sensitive to self-supervised pre-training of student. We conjecture that since Flowers102 dataset has only 1020 training samples for the 102 classes, it is not enough to probe the teacher's representation space.

**Qualitative result.** To see the quality of the transferred representations, we analysed text-image retrieval results on the Flowers102 dataset. Figure 3 compares the top-5 text-image retrieval results between CLIP-RN50 and BeamCLIP-RN50. A red rectangle denotes an incorrect result. Compared to CLIP-RN50, BeamCLIP-RN50 provides much improved results, since its representations are transferred from CLIP-ViT/16 with higher zero-shot accuracy. More interestingly, BeamCLIP-RN50 provides surprisingly good text-image retrieval results, even though unseen text prompts such as "a photo of {pink rose}" or "a photo of {yellow rose}" are given.

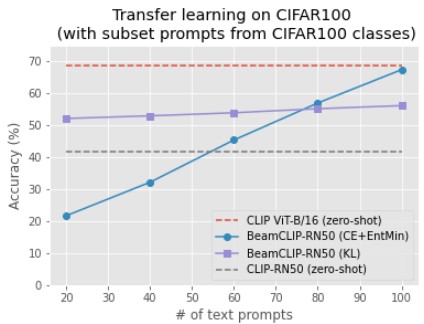
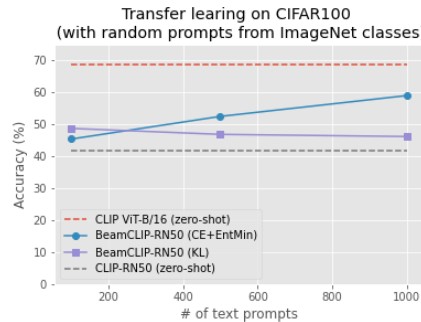

(a) Subset text prompts from CIFAR100 classes        (b) Random text prompts from ImageNet classes

Figure 4: **The effect of random text prompts on CIFAR100.** **(a)** The text prompts are randomly sampled from the set of 100 class names of CIFAR100. The red dotted line denotes the teacher's accuracy as an upper bound. It is more efficient as it is closer to this line. As shown in the blue line, the BeamCLIP (CE+EntMin) can effectively transfer the CLIP representations, even when the class names of the target dataset are partially given. **(b)** The text prompts are randomly sampled from the 1000 class names of ImageNet. The BeamCLIP (CE+EntMin) is still effective, even though the class names are randomly sampled from a non-target dataset (ImageNet-1K).

## 4.4    Effect of random text prompts

We measured how effective the BeamCLIP is in cases where the class names of the target dataset are not perfectly given. Figure 4 shows the effect of the randomly sampled text prompts on CIFAR100. We can see that the BeamCLIP is still effective, even when (a) the subset of the 100 class names of CIFAR100 are given as the text prompts, or (b) the text prompts are randomly sampled from a non-target dataset (ImageNet-1K). The exact values in Figure 4 are presented in Table 15 and Table 16 in Appendix C.2. Also, the additional results on CIFAR10 are provided in Appendix C.2.

## 5    Limitations and Conclusion

**Limitations.** With the help of rich representations of pre-trained CLIP, the BeamCLIP can learn better representations than SSL methods. However, since SSL methods can increase the performance at longer training epochs, the performance margin may be decreased in such a setting. Another shortcoming is that context-based prompt augmentations may require additional engineering efforts.

**Conclusion.** In this paper, we provide the BeamCLIP that can effectively transfer large pre-trained vision-language model (e.g., CLIP-ViT) into a small target model (e.g., ResNet-18) with cross-modal similarity matching (CSM) and context-based prompt augmentation (CPA). We empirically show that the BeamCLIP can learn better visual representations than vision-only self-supervised learning (SSL) methods, by leveraging a pre-trained vision-language model (CLIP). The BeamCLIP is not intended to be another CLIP, but an effective CLIP student.

## Broader impact

This research aims to provide a simple and effective way to leverage CLIP for representation learning. With the help of CLIP, the BeamCLIP can learn better representations than self-supervised learning (SSL) methods. Since training CLIP requires very large data and hundreds of GPUs, it is important to provide a way to effectively reuse the pre-trained CLIP rather than training from scratch on a target model. We believe that the BeamCLIP can help to save cost and time.

## Acknowledgements

We thank anonymous reviewers for their valuable comments. This work was fully supported by LG AI Research.

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
