# A  Datasets Details

We demonstrate the effectiveness of the BeamCLIP by using six downstream datasets. Table 8 shows the details of the downstream datasets.

Table 8: Details of datasets used for the BeamCLIP evaluation.

| Dataset | Image Size | Classes | Train Size | Val Size | Test Size |
|---|---|---|---|---|---|
| CIFAR10 [23] | 32x32 | 10 | 40,000 | 10,000 | 10,000 |
| CIFAR100 [23] | 32x32 | 100 | 40,000 | 10,000 | 10,000 |
| STL10 [9] | 128x128 | 10 | 4,000 | 1,000 | 8,000 |
| Flowers102 [28] | 224x224 | 102 | 1,020 | 1,020 | 6,149 |
| Pets37 [31] | 224x224 | 37 | 2,944 | 736 | 3,669 |
| ImageNet [10] | 224x224 | 1,000 | 1,231,167 | 50,000 | 50,000 |

# B  Method Details

In this section, we provide some details of the BeamCLIP . More specifically, we provide the details of two main contributions that are (1) cross-modal similarity matching (CSM) and (2) context-based prompt augmentation (CPA). Also, we provide the other implementation details such as image augmentation, similarity smoothing, model hyperparameters, etc.

## B.1  Image augmentation details

We use conventional image augmentation when performing representation transfer by using unlabeled images in downstream datasets. Table 9 provides a list of image augmentation used for unsupervised representation transfer on downstream datasets.

Table 9: A list of image augmentations used in the BeamCLIP .

| Mode | Augmentation | Parameters |
|---|---|---|
| Train | RandomResizedCrop | - |
| | RandomHorizontalFlip | p=0.5 |
| | RandomColorJitter | p=0.8 |
| | GaussianBlur | p=0.5, min=0.1, miax=2.0 |
| | Normalize | - |
| Val | Resize | input_size + 0.1 * input_size |
| | CenterCrop | input_size |
| | Normalize | - |

## B.2  Cross-modal similarity matching details

Cross-modal similarity matching (CSM) is the main method of the BeamCLIP . To make the concept of CSM clearer, we provide an illustration of CSM in Figure 5.

## B.3  Context-based prompt augmentation details

To prepare for better text anchor embeddings for unsupervised representation transfer, we introduce context-based prompt augmentation (CPA). To make the concept of CPA clearer, we provide an illustration of CPA in Figure 6.

Also, we provide an example of the hierarchical class labels in Table 10 and an example context-based prompt augmentation for CIFAR100 in Table 11.

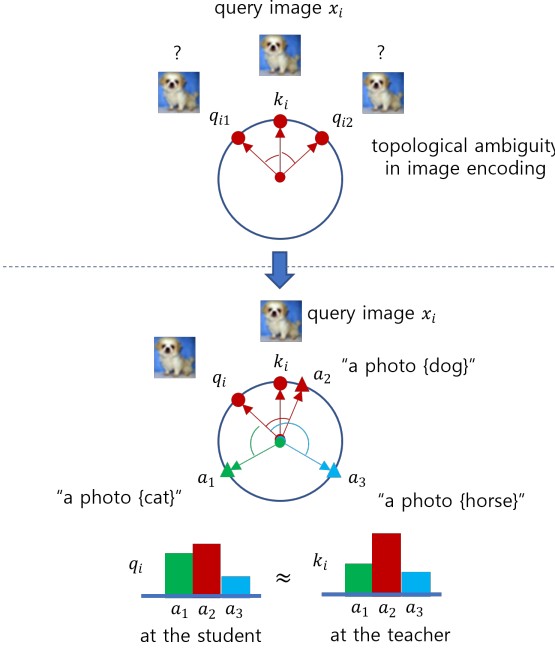

Figure 5: **Illustration of cross-modal similarity matching.** Topological ambiguity may occur in image encoding, since query image embedding $q_{i1}$ and $q_{i2}$ can have the same cosine similarity compared to a single teacher image embedding $k$, while heading towards different directions. To mitigate this problem, we introduce cross-modal similarity matching that encourage the student to mimic the same cross-modal similarity distribution (measured against multiple anchor text points) in teacher's embedding space.

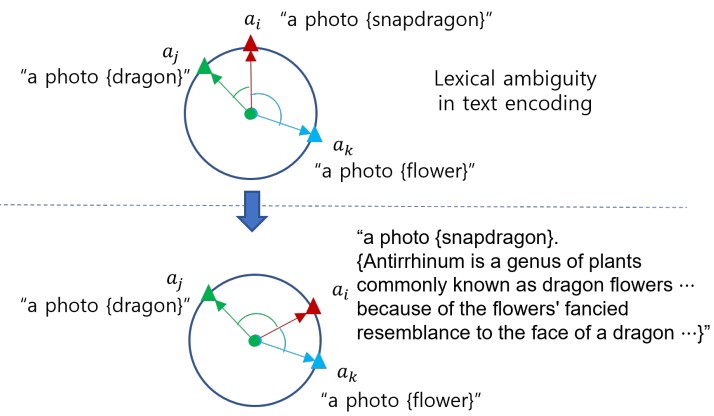

Figure 6: **Illustration of context-based prompt augmentation.** The lexical ambiguity may occur in text encoding, since the same text may have multiple different meanings. To mitigate this problem, we introduce context-based prompt augmentation that helps resolve the ambiguity with contextual texts such as Wikipedia descriptions.

### B.4   Other implementation details

**Self-supervised pre-training of student.**   For self-supervised pre-training, we adopt SimCLR, since it is simple and effective. SimCLR learns transferable visual representations by using InfoNCE loss

Table 10: Coarse and fine labels for CIFAR100.

| Coarse Label | Fine Label |
|---|---|
| aquatic mammals | beaver, dolphin, otter, seal, whale |
| fish | aquarium fish, flatfish, ray, shark, trout |
| flowers | orchid, poppy, rose, sunflower, tulip |
| food containers | bottle, bowl, can, cup, plate |
| household electrical devices | clock, keyboard, lamp, telephone, television |
| household furniture | bed, chair, couch, table, wardrobe |
| insects | bee, beetle, butterfly, caterpillar, cockroach |
| large carnivores | bear, leopard, lion, tiger, wolf |
| large man-made outdoor things | bridge, castle, house, road, skyscraper |
| large natural outdoor scenes | cloud, forest, mountain, plain, sea |
| large omnivores and herbivores | camel, cattle, chimpanzee, elephant, kangaroo |
| medium mammals | fox, porcupine, possum, raccoon, skunk |
| non-insect invertebrates | crab, lobster, snail, spider, worm |
| people | baby, boy, girl, man, woman |
| reptiles | crocodile, dinosaur, lizard, snake, turtle |
| small mammals | hamster, mouse, rabbit, shrew, squirrel |
| trees | maple tree, oak tree, palm tree, pine tree, willow tree |
| vehicles 1 | bicycle, bus, motocycle, pickup truck, train |
| vehicles 2 | lawn mower, rocket, streetcar, tank, tractor |

Table 11: Examples of prompt augmentation with hierarchical labels for CIFAR100.

| Label Name | Text Prompt |
|---|---|
| baby | "A photo of a {baby}, categorized as {people}." |
| beaver | "A photo of a {beaver}, categorized as {aquatic mammals}." |
| bee | "A photo of a {bee}, categorized as {insect}." |

[30, 39] which encourages agreement between multiple views of the same image. More specifically, InfoNCE maximizes the similarity between multiple views of the same image (*i.e.,* positive samples) and minimizes the similarity to multiple views of all other images in a training batch (*i.e.,* negative samples). InfoNCE loss of SimCLR can be formulated as follows:

$$\mathcal{L}_{\text{InfoNCE}} = -\log \frac{\exp\left((h_i^S \cdot h_{i'}^S)/\tau\right)}{\sum_{k=1}^{2B} \mathbb{1}_{[k \neq i]} \exp\left((h_i^S \cdot h_k^S)/\tau\right)} \tag{10}$$

where $h_i^S \in \mathbb{R}^{128}$ is a projection of a student representation $q_i \in \mathbb{R}^{512}$, $\tau$ is a temperature hyperparameter that is set to 0.1, $\mathbb{1}_{[k \neq i]}$ is an indicator function whose value is 1 if $k \neq i$, and $B$ is a batch size. Here, $h_i$ and $h_{i'}$ are projections of multiple views of the same input images $x_i$.

**Similarity Smoothing.** To improve the effectiveness of distillation, we apply Label Smoothing (LS) [36] to the cross-modal similarity distillation loss. Recent works [27, 42] show that Label Smoothing helps knowledge distillation. To apply Label Smoothing, we determine the most similar anchor representation as follows:

$$j^* = \arg\max_j s_j(k_i, A). \tag{11}$$

Then, we generate a modified cross-modal similarity distribution:

$$s_j(k_i, A)^{LS} = \mathbb{1}_{[j=j^*]}(1 - \alpha) + \alpha/M \tag{12}$$

where $\mathbb{1}_{[j=j^*]}$ is the indicator function whose value is 1 if $j = j^*$, $M$ is the number of anchors, and $\alpha$ is the smoothing hyperparameter that is set to 0.2 in our experiments.

## B.5 Model hyperparameters

Table 12 provides the summary of model hyperparamters. We use the same hyperparameters on all downstream datasets if not explicitly declared.

Table 12: BeamCLIP hyperparameters.

| Hyperparameter | Value |
|---|---|
| CSM loss temperature | 0.01 |
| ISM loss scale | {0.1, 1.0, 10.0} |
| similarity smoothing (LS) sacle | 0.2 |
| optimizer | SGDR [25] |
| initial learning rate | 0.5 |
| weight decay | 1e-6 |
| EMA momentum | 0.99 |
| batch size | {256, 512} |
| epochs | 200 |

## C  Additional Experiment Results

In this section, we provide additional experiment results. First, we provide the learning curves that are generated while training the BeamCLIP . Second, we provide some experiment results on the effects of random text prompts. Third, we provide an example qualitative result that shows the advantage of the BeamCLIP .

### C.1  Learning curves of the BeamCLIP

We provide the learning curve of the BeamCLIP for the experiment section. Figure 7 shows the learning curve for ImageNet-1K validation accuracy of BeamCLIP-RN50 representations trained with unlabeled ImageNet-1K. Figure 8 shows the learning curve for ImageNet-1K validation accuracy of BeamCLIP-RN18 representations trained with unlabeled ImageNet-1K. Figure 9 shows the learning curve for ImageNet-1K zero-shot accuracy of BeamCLIP-RN50 representations trained with unlabeled non-target data (ImageNet-21K).

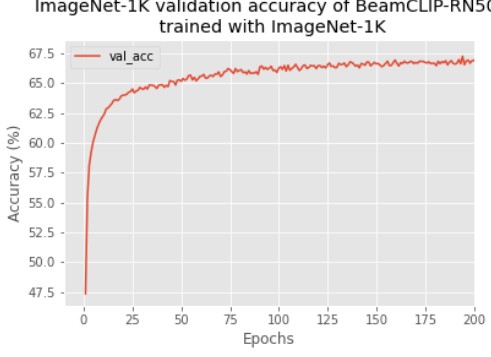

Figure 7: ImageNet-1K top-1 validation accuracy of BeamCLIP-RN50 representations learned with unlabeled target data (ImageNet-1K).

### C.2  The effect of random text prompts

In this section, we further analyze the effect of text prompts from the perspective of unsupervised learning. Before that, we briefly review the proposed method. In this paper, we propose the BeamCLIP , an unsupervised representation transfer method of a large pre-trained multimodal model such as CLIP. The BeamCLIP can transfer the visual representations of CLIP by using unlabeled images on a downstream dataset. To achieve this, we propose cross-modal similarity matching (CSM). In CSM, at first, given an unlabeled image, cross-modal similarity distribution is measured from multiple text prompt embeddings in the teacher's embedding space. Then, a student model is encouraged to mimic the cross-modal similarity distribution of the teacher model by matching these similarity distributions. To achieve effective transfer, we use anchor text embeddings by encoding text prompts. For example,

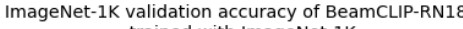
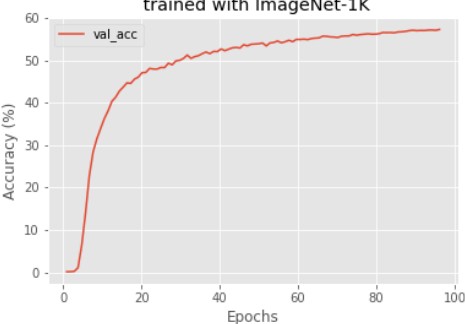

Figure 8: ImageNet-1K top-1 validation accuracy of BeamCLIP-RN18 representations learned with unlabeled target data (ImageNet-1K).

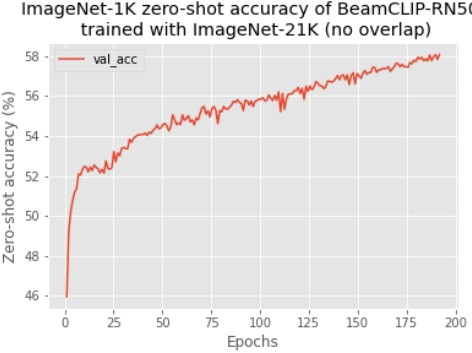

Figure 9: Zero-shot ImageNet-1K top-1 accuracy of BeamCLIP-RN50 representations learned with unlabeled non-target data (ImageNet-21K).

on CIFAR10, we use ten text prompts in the form of `"a photo of {class name}"`. Note that the text prompts are not paired with each image.

**CIFAR10.** We measured how effective the BeamCLIP is in cases where the class names of the target dataset are not perfectly given. Table 10 shows the effect of the randomly sampled text prompts on CIFAR10 [23]. The values in Figure 10 are also presented in Table 13 and Table 14.

**CIFAR100.** Table 4 shows the effect of the randomly sampled text prompts on CIFAR100 [23]. The values in Figure 4 are also presented in Table 15 and Table 16.

Table 13: Effect of the partial text prompts on CIFAR10.

| Method | Type | Img. Enc. | Prompts | | | | | - |
| | | | 3 | 5 | 7 | 9 | 10 | |
| --- | --- | --- | --- | --- | --- | --- | --- | --- |
| CLIP [32] (zero-shot) | T | ViT-B/16 | - | - | - | - | - | 91.6 |
| CLIP [32] (zero-shot) | - | RN50 | - | - | - | - | - | 75.6 |
| BeamCLIP (CE+EntMin) | S | RN50 | 83.47 | 83.25 | 88.26 | 91.84 | **92.10**$^{*}$ | - |
| BeamCLIP (KL) | S | RN50 | 89.36 | 90.43 | 90.15 | 90.54 | 90.85 | - |

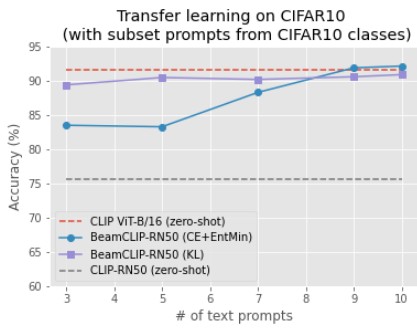
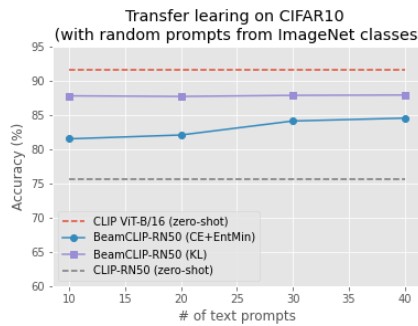

(a) Subset prompts from CIFAR10        (b) Random prompts from ImageNet

Figure 10: **Effect of the random text prompts on CIFAR10. (a)** The text prompts are randomly sampled from the 10 class names of CIFAR10. The red dotted line denotes the teacher's accuracy as an upper bound. It is more efficient as it is closer to this line. As shown in the blue line, the BeamCLIP is still effective, even when the class names of the target dataset are partially given. The BeamCLIP (KL) means to use the KL-divergence for matching cross-modal similarity distribution. The BeamCLIP (CE+EntMin) is more effective, as more text prompts are given. **(b)** The text prompts are randomly selected from the 1000 class names of ImageNet. The BeamCLIP (CE+EntMin) is still effective, even though the class names are randomly sampled from a non-target dataset (ImageNet-1K).

Table 14: Effect of the random text prompts on CIFAR10.

| Method | Type | Img. Enc. | 10 | 20 | 30 | 40 | - |
|---|---|---|---|---|---|---|---|
| | | | | Prompts | | | |
| CLIP [32] (zero-shot) | T | ViT-B/16 | - | - | - | - | **91.6** |
| CLIP [32] (zero-shot) | - | RN50 | - | - | - | - | 75.6 |
| BeamCLIP (CE+EntMin) | S | RN50 | 81.49 | 82.05 | 84.09 | **84.51** | - |
| BeamCLIP (KL) | S | RN50 | 87.76 | 87.67 | 87.83 | 87.88 | - |

Table 15: Effect of the partial text prompts on CIFAR100.

| Method | Type | Img. Enc. | 20 | 40 | 60 | 80 | 100 | - |
|---|---|---|---|---|---|---|---|---|
| | | | | | Prompts | | | |
| CLIP [32] (zero-shot) | T | ViT-B/16 | - | - | - | - | - | **68.7** |
| CLIP [32] (zero-shot) | - | RN50 | - | - | - | - | - | 41.6 |
| BeamCLIP (CE+EntMin) | S | RN50 | 21.72 | 32.19 | 45.38 | 56.93 | **67.35** | - |
| BeamCLIP (KL) | S | RN50 | 52.10 | 52.93 | 53.88 | 55.15 | 56.12 | - |

Table 16: Effect of the random text prompts on CIFAR100.

| Method | Type | Img. Enc. | 100 | 500 | 1000 | - |
|---|---|---|---|---|---|---|
| | | | | Prompts | | |
| CLIP [32] (zero-shot) | T | ViT-B/16 | - | - | - | **68.7** |
| CLIP [32] (zero-shot) | - | RN50 | - | - | - | 41.6 |
| BeamCLIP (CE+EntMin) | S | RN50 | 45.36 | 52.44 | **58.93** | - |
| BeamCLIP (KL) | S | RN50 | 48.68 | 46.82 | 46.15 | - |