# OpenReview forum: "Transferring Pre-trained Multimodal Representations with Cross-modal Similarity Matching"
_NeurIPS.cc/2022/Conference — NeurIPS 2022 Accept_

### Official Review · Reviewer_zQJm · 2022-06-27

**Rating:** 5
**Confidence:** 4
**Soundness:** 3 good
**Presentation:** 3 good
**Contribution:** 3 good

**Summary:**

The authors propose a method for transferring CLIP's visual
representations into smaller models; this setup might be useful for
researchers working on, e.g., low resource devices, who can't run the
more powerful, full CLIP models. Their method not only minimizes the
cosine similarity between the image representations of a teacher
(ViT-B/16) and student (RN50) model, but also, incorporates a set of
"anchor" vectors, computed by the text side of CLIP: the cross-modal
similarity objective encourages the cross-modal similarities between
the teacher and the student to match, too.

**Questions:**

- Did the authors try transferring to datasets where the label set and the images are not available at training time?

- Did the authors try with different teachers/students?

**Limitations:**

The authors mainly focus on making higher performance, lower-resource versions of existing CLIP models. Given that CLIP's negative impacts have been discussed in the original work and follow ups, I don't think significant additional discussion is needed here.

I did think that the "Broader impacts" section was a bit short: can more information about FLOPs/energy use/on-device computation be discussed?

**Strengths And Weaknesses:**

Strengths:

The work is easy to follow, and save for a few typos (see below), easy
to follow. The empirical results are promising: the authors
successfully distill ViT-B/16-CLIP into a set of RN50 weights. The
cross-modal matching objective is clever, and thorough ablations
demonstrate that this innovation is indeed complementary (and, in fact,
in isolation, is actually *more* important than the cosine loss!). I
appreciated that the authors evaluated on a number of datasets beyond
ImageNet, albeit only in a zero-shot capacity.

Weaknesses:

- The authors distill a ViT-B/16 into a RN50. While the former takes
  more FLOPs than the later, ViT-B/32 is cheaper than RN50, and the
  CLIP version of that architecture outperforms both the author's RN50
  and the CLIP RN50 model.  While this doesn't totally undercut the
  presented results, it's a bit strange to me why the authors chose
  the particular distillation pair that they did --- it seems like
  ViT-B/32 (or even EfficientNet or MobileNet) would have been a good
  choice for student, and perhaps L/14-336px CLIP as the teacher would
  have made a more compelling setup.

- The ablations suggest that the cosine similarity loss isn't really
  required: the best zero-shot imagenet performance is actually
  achieved only with the cross-modal loss, and the other datasets seem
  to be within a close margin. This isn't really a /negative/,
  per-say, but a bit contrary to the story told.

- It would have been nice to see linear probe results for the other
  datasets like Pets37 as well.

- Even assuming that RN50 is the best choice of student, the empirical
  results are somewhat unimpressive. Specifically, I believe that the
  authors choose anchor points to be task-specific prompts and the author's
  model is domain adapted to (unlabelled) dataset-specific images, i.e.,
  there's reason to believe that the author's method is specifically
  tuned to the tasks described. RN50-CLIP, which is not tuned to the
  specific tasks described, achieves only slightly worse imagenet linear
  probe accuracy: 73.3 vs 74.8.

- The zero shot results in table 3 are interesting, but a bit
  incomplete. The ViT-B/32-CLIP model smaller than the author's
  distilled RN50 model achieves only slightly worse performance on
  these tasks, but it isn't presented. Again, this observation doesn't
  invalidate the author's results, it's just that one really needs to
  buy that RN50 specifically is an interesting student model.

- Overall, the authors clearly "win" on this particular architecture
  by a few accuracy points on linear probe imagenet, but it's not
  clear that this is really the best architecture to distill to
  (particularly because ViT-B/32 is smaller and there's already a CLIP
  model for that). To the author's credit, they recognize this on
  L288, but I would have preferred to see that the author's method
  generalizes to other teacher/student models, instead of just this
  one combination.

Overall, the authors have a promising core result: it's possible to
distill the knowledge of a CLIP model into a smaller set of
weights. And, the innovation that enables this distillation (that
image-text distances should be preserved in addition to image-image
distances) is novel. However, the empirical results are slightly
underwhelming: the authors model, by virtue of the selection of the
anchor points and distillation over unlabelled images from the corpora
of interest, is tuned to this particular set of datasets (vs. CLIP
that isn't). Their performance improvement over CLIP is somewhat small
in magnitude (73.3 vs 74.8). I would have liked to have seen: 1)
distillation to significantly more efficient architectures vs. RN50
(ViT-B/32 CLIP is already smaller) and with better teachers than
ViT-B/16; and 2) linear probe evaluations on unseen tasks, where the
images haven't been seen by the distillation model at training time.




Presentation fixes:

- It should be specified that Table 3 results are *all* zero shot.
- L71: week --> weak
- The broader impact statement is a bit hollow -- I would have appreciated a fuller discussion of embedded systems and/or low resource computation
- Table 4's caption has an incomplete sentence.

---

> ### Author Response · Authors · 2022-08-02
> **We thank the reviewer for his/her comments and suggestions.**
>
> ***"Beam CLIP Up, Scotty." -- Captain Kirk in Star Trek --***
>
> **Q1.** Did the authors try transferring to datasets where the label set and the images are not available at training time?
>
> **A1.** To answer this important question, we have added **Table 5 "Comparison of zero-shot accuracy on ImageNet-1K"** in the revised paper. To check if the BeamCLIP also inherits the powerful zero-shot capability of CLIP, we compare zero-shot accuracy of CLIP variants on ImageNet-1K. For zero-shot measure, we use CC-3M and ImageNet-21K (12M samples) that do not overlap with ImageNet-1K (i.e., ImageNet-20K). ***The BeamCLIP-RN50 achieves about 57.5% zero-shot accuracy that is comparable with CLIP RN-50 (59.6%).*** To achieve such a high zero-shot accuracy, CLIP uses very large image-text pair data (WIT-400M). Instead, the BeamCLIP can achieve the comparable zero-shot accuracy by effectively transferring the teacher's representations, while using only 3% data (ImageNet-21K (12M)). ***We note that CLIP and OpenCLIP [1] provide only about 31.3% and 32.7% zero-shot accuracy with the similar amount of data (YFCC-15M).*** We carefully expect that if we can use larger training data (e.g., about 10~20% of WIT-400M), we may reach or surpass 59.6% accuracy of CLIP RN-50 using WIT-400M. By the way, ***the BeamCLIP is not intended to be an another CLIP, but an effective CLIP student.***
>
> | Method | Image Encoder | Training Data   | Teacher Model | Text Prompts | Batch Size  | Epochs | ImageNet Zero-shot |
> |-------------------------|--------|----------------|---|---|----------|----|-------|
> | CLIP                          | RN50 | WIT-400M      | x | x | 32,768  | 32 | 59.6 |
> | CLIP                          | RN50 | YFCC-15M    | x | x | -            | -   | **31.3** |
> | OpenCLIP [1]            | RN50 | YFCC-15M     | x | x | 256 * 8  | 32 | 32.7 |
> | OpenCLIP [1]            | RN50 | CC-12M         | x | x | 256 * 8  | 32 | 36.5 |
> | **BeamCLIP** (ours) | RN50 | CC-3M           | CLIP ViT-B/16 | IN-1K | 64 * 8 | 100 | 49.5 |
> | **BeamCLIP** (ours) | RN50 | IN-21K (12M) | CLIP ViT-B/16 | IN-1K | 64 * 8 | 190 | **57.5** |
>
> ***
> **Q2.** Did the authors try with different teachers/students?
>
> **A2.** To check if the BeamCLIP can transfer CLIP representations into smaller models than ResNet-50 (24M), we also measure ImageNet-1K Top-1 linear probe accuracy on ResNet-18 (11M). ResNet-18 is trained from scratch (not self-supervised pre-trained with SimCLR), while transferring CLIP ViT-B/16 representations. **Figure 1 and Table 3** in the revised paper shows **ImageNet-1K top-1 linear probe accuracy on ResNet-18 representations**. Similar to ResNet-50, BeamCLIP learns better representations than SSL methods such as SimCLR, MoCo-v2, BYOL, and SwAV. More importantly, the BeamCLIP provides better performance than OSS [1] that simultaneously learns and transfers representations from ResNet-50.
>
> | Method      | Teacher | Student | 100 | 200 | 400 |
> |--------------|---|--------|-------|-------|------|
> | Supervised | x | RN18 |           69.8           |
> | SimCLR      | x | RN18 | 47.1 | 49.9 | 51.8 |
> | MoCo-v2    | x | RN18 | 48.6 | 49.9 | 51.9 |
> | BYOL          | x | RN18 | 44.2 | 47.5 | 46.8 |
> | SwAV          | x | RN18 | 57.7 | 61.2 | 63.7 |
> | OSS [2]       | SSL RN50 | RN18 | 60.0 | 64.1 | 65.8 |
> | **BeamCLIP** (ours) | CLIP ViT-B/16 | RN18 | **63.8** | **64.8** | **66.2** |
>
> ***
> **Q3.** I did think that the "Broader impacts" section was a bit short: can more information about FLOPs/energy use/on-device computation be discussed?
>
> **A3.** We have added the “Limitation” paragraph in the “Conclusion and Limitations” section. We mentioned two issues: performance gain at long training epochs and additional effort for context-based prompt augmentation (CPA). Also, we have revised the “Broader impacts” section.
>
> This research aims to provide a simple and effective way to leverage CLIP for representation learning. With the help of CLIP, the BeamCLIP can learn better representations than self-supervised learning (SSL) methods. Since training CLIP requires very large data and hundreds of GPUs, it is important to provide a way to effectively reuse the pre-trained CLIP rather than training from scratch on a target model. We believe that BeamCLIP can help to save cost and time.
>
> ***
> **References**
> [1] G. Ilharco, M. Wortsman, R. Wightman, C. Gordon, N. Carlini, R. Taori, A. Dave, V. Shankar,353
> H. Namkoong, J. Miller, H. Hajishirzi, A. Farhadi, and L. Schmidt. Openclip, jul 2021. https://github.com/mlfoundations/open_clip
> [2] H. M. Choi, H. Kang, and D. Oh. Unsupervised representation transfer for small networks: I believe I can distill on-the-fly. In Advances in Neural Information Processing Systems (NeurIPS), 2021.

---

> > ### Comment · Reviewer_zQJm · 2022-08-08
> > **Thanks!**
> >
> > Hi Authors,
> >
> > Thanks for your thoughtful reply, and for the Star Trek quote. Currently, openreview's pdf serving appears to be down, so I plan to circle back more fully during the additional review phase. But, I will do my best with the cached version of your submission I have and your comment.
> >
> > Re: additional experiments with different training data --- my main concern was that the distillation process has additional supervision vs. the true zero-shot case because it's fine-tuned with knowledge of the class names come test time. While the additional experiments you ran are quite interesting and provide perspective on a related question (i.e., what if the images are from a different dataset), I'm curious if/how these additional results address my original concern.
> >
> > Re: ResNet18 experiments --- thanks for these, this is great! I raised my score by a point for these new experiments.

---

> > > ### Author Response · Authors · 2022-08-09
> > > **We thank the reviewer for his/her kind response.**
> > >
> > > We really appreciate your comments and suggestions. With your valuable comments, we could further enhance our paper. We have carefully revised our paper and uploaded it to OpenReview. We highlighted revised parts with the blue color.
> > >
> > > Regarding the training data, we categorized the setting into two: (1) representation transfer with unlabeled target data (i.e., ImageNet-1K) and (2) representation transfer with unlabeled non-target data (e.g., ImageNet-21K or CC-3M). The later setting may or may not have class names for a target dataset. In the revised paper, we have provided a result with target class names given. However, the BeamCLIP can learn representations with random class names (i.e., random text prompts). In this setting, the accuracy (zero-shot or linear probe) may slightly decrease compared to the setting with exact class names given. Related experiments have been provided in **Figure 8 and 9 of Appendix C.2 “Effect of random text prompts”**. Please examine the result whether they are related to your additional comment.

---

### Official Review · Reviewer_eofu · 2022-06-29

**Rating:** 7
**Confidence:** 4
**Soundness:** 3 good
**Presentation:** 4 excellent
**Contribution:** 3 good

**Summary:**

This paper aims at transferring knowledge from a pre-trained language-and-vision model to a small CNN-based vision network. The authors argue that a smaller network can learn better from multimodal representations in comparison with previous vision-only self-supervised methods. For transferring knowledge they propose to use two similarity matching objectives: one that matches visual representations from the teacher and student networks, and one (cross-modal) that matches the visual representations of the student network with the textual representations from CLIP. In order to increase robustness on the cross-modal similarity matching, the authors further propose to use multiple anchor textual representations, and they better craft textual prompts depending on the task and dataset. Finally, they first pre-train the student network based on previous self-supervised objectives, before distilling knowledge from CLIP. Experimentally, they show that the representations transfer well to image classification tasks, they are competitive against supervised methods and improve over self-supervised vision-only approaches.

**Questions:**

1. Have you tried using other self-supervised methods for pre-training except for SimCLR?
2. Why specific datasets (e.g., CIFAR100, Flowers102; Table 4) are so sensitive depending on pre-training in comparison with others (e.g., ImageNet)?


**Limitations:**

As mentioned in the weaknesses of the paper, there are two limitations: 1. The fact that the authors did not use multiple self-supervised pre-training methods and/or one of the most competitive (e.g., BYOL) for drawing safe conclusions given the large impact that pre-training has, and 2. There is no further analysis on the pre-training factor, which is so important for good performance.

**Strengths And Weaknesses:**

**Strengths**

1. The paper presents an interesting approach with novel components (e.g., cross-modal similarity matching, augmented textual representations) for transferring knowledge from multimodal representations to smaller vision-only encoders.
2. The authors experimentally demonstrate that their approach offers good visual representations that improve accuracy on image classification over prior self-supervised vision-only methods. Moreover, the performance is now closer to supervised methods, closing the gap further.


**Weaknesses**

1. From the experimental results, it is demonstrated that pre-training with prior self-supervised objectives before distilling knowledge from CLIP is crucial for performance, especially in certain datasets. However, the authors have chosen the weakest self-supervised model (SimCLR; Table 2) for pre-training. Results and conclusions would be more complete if they have chosen more than one methods for pre-training and/or one the strongest ones. Now, there is a question of whether the same patterns can be found independently of the pre-training methods and whether specific of these methods have advantage as a pre-training step for knowledge distillation from CLIP.
2. As mentioned in the above point, self-supervised pre-training is crucial, and for specific datasets (e.g., CIFAR100, Flowers102; Table 4) not pre-training has an overwhelming performance degradation (e.g., 12.94% accuracy vs. 62.18%). Given these huge gaps and the necessity of pre-training for the method to work, further analysis is needed and some discussion about why specific datasets are more sensitive to this than others.

---

> ### Author Response · Authors · 2022-08-02
> **We thank the reviewer for his/her comments and suggestions.**
>
> **Q1.** Have you tried using other self-supervised methods for pre-training except for SimCLR?
>
> **A1.** We selected SimCLR for two main reasons. First, we want to minimize the effect of pre-training of the student model. Second, we want to reduce the dependency to some unique modules. Instead, SimCLR is simple and widely used.
>
> To answer this important question, we have added **Table 4 "Effect of self-supervised pre-training of the student"** in the revised paper. Table 4 shows linear probe accuracy on BeamCLIP-RN50 representations by using different SSL pre-training. ***With a better SSL method (e.g., SwAV), the BeamCLIP can learn better representations with an increased linear probe accuracy.*** Also, the gap between supervised and unsupervised is considerably reduced (76.2% vs. 75.8%). We carefully expect that if we train the model for longer epochs, we may achieve an increased linear probe accuracy. Even though self-supervised pre-training of a student helps the BeamCLIP to learn better representations, we would like to highlight that the BeamCLIP is not dependent on it. For example, in case of transfer learning on ImageNet-1K, the BeamCLIP can achieve high accuracy without self-supervised pre-training of the student (w/o SSL PT: 63.30% vs. w/ SSL PT: 65.45%).
>
> | Method | Teacher | Student | Pre-training of Student | Batch | Epoch | Linear Probe |
> |----------|-----------|----------|----------------------------|--------|---------|----------------|
> | Supervised | x | RN50 | x | 256 | - | 76.2 |
> | **BeamCLIP** (ours) | CLIP ViT-B/16 | RN50 | SimCLR | 512 | 200 | 74.8 |
> | **BeamCLIP** (ours) | CLIP ViT-B/16 | RN50 | SwAV | 512 | 100 | **75.8** |
>
> ***
> **Q2.** Why specific datasets (e.g., CIFAR100, Flowers102; Table 4) are so sensitive depending on pre-training in comparison with others (e.g., ImageNet)?
>
> **A2.** Flowers102 dataset has only 1020 training samples for 102 classes (i.e., ten samples for each class) We conjecture that this small number of samples is not enough to probe the teacher's representation space. We have added some analysis on this in the **"Ablation study" paragraph of the Section 4.3** in the revised paper.

---

### Official Review · Reviewer_nz7y · 2022-07-11

**Rating:** 6
**Confidence:** 3
**Soundness:** 3 good
**Presentation:** 3 good
**Contribution:** 3 good

**Summary:**

The authors propose a method for transferring the representation power of CILP ViT-B/16 to a small ResNet-50 by utilizing the cross-modality of the embedding space.
Specifically, in the training stage, a student model not just tries to mimic the output image features of a teacher model, but also tries to follow the similarity distribution of the teacher model that is computed with the given set of text descriptions.
These text descriptions are similar to the ones that were used in the zero-shot classification task in the original CLIP paper, but it is amplified with the additional information of each image that could be found on Wikipedia.
Using this additional cross-modal guidance, the proposed method was able to train student networks that presents the SOTA performance on multiple classification tasks.

**Questions:**

Questions
- From Table 4, I could see that SSL PT plays an important factor for some cases, though it is not described in the method part. Could you explain why this factor has such huge effect in the results?

Suggestions
- From Table 4, I found that CPA is most helpful for dataset Flowers102, which implies that this method is especially effective for fine-grained datasets. It was interesting for me, and I think tackling this kind of dataset-wise differences in effectiveness of each element would be great for following research.
- Guess Figure 3 is plotting the same figure for two models?

**Limitations:**

Authors did not address the limitations and potential negative societal impact of their work.

Suggestions
- I want to know the authors’ opinion on how much the transfer learning is helpful for handling issues concerned with sensitive web data?

**Strengths And Weaknesses:**

Strengths
- The authors propose the novel technique to guide a student model using the multi-modality of CLIP embedding space.
- The proposed method proved to be effective in boosting classification performance on multiple datasets.

Weaknesses
- The analysis on the experimental results is somewhat scant. It could be amplified by providing the dataset-wise differences and more results from the larger number of prior transferring methods.

---

> ### Author Response · Authors · 2022-08-02
> **We thank the reviewer for his/her comments and suggestions.**
>
> **Q1.** From Table 4, I could see that SSL PT plays an important factor for some case, though it is not described in the method part. Could you explain why this factor has such huge effect in the results?
>
> **A1.** Flowers102 dataset has only 1020 training samples for 102 classes (i.e., ten samples for each class) We conjecture that this small number of samples is not enough to probe the teacher's representation space. We have added some analysis on this in the "Ablation study" paragraph of the Section 4.3 in the revised paper.
>
> ***
> **Q2.** From Table 4, I found that CPA is most helpful for dataset Flowers102, which implies that this method is especially effective for fine-grained datasets. It was interesting for me, and I think tackling this kind of dataset-wise differences in effectiveness of each element would be great for following research.
>
> **A2.** Thank you for your insightful comments. We agree that ***CPA is most helpful for fine-grained datasets***. We also think that the dataset-wise difference needs to be further analyzed.
>
> ***
> **Q3.** Guess Figure 3 is plotting the same figure for two models?
>
> **A3.** We have revised Figure 3 to show different images from each model. By the way, we have moved it into **Figure 10 of Appendix C.3 ("Qualitative results")** in the revised paper, since we need to place additional experiment results in the experiment section.
>
> ***
> **Q4.** Authors did not address the limitations and potential negative societal impact of their work.
>
> **A4.** We have added the “Limitation” paragraph in the “Conclusion and Limitations” section. We mentioned two issues: performance gain at long training epochs and additional effort for context-based prompt augmentation (CPA).

---

### Official Review · Reviewer_izRQ · 2022-07-12

**Rating:** 3
**Confidence:** 4
**Soundness:** 2 fair
**Presentation:** 3 good
**Contribution:** 2 fair

**Summary:**

The authors propose a method (BeamCLIP) that can effectively transfer the representations of a large pre-trained multimodal model (CLIP ViTB/16) into a small target model (ResNet-50). The authors introduce two main components for the representation transfer: cross-modal similarity matching (CSM) and context-based prompt augmentation (CPA). CSM enables a student model to learn the representations of a teacher model by matching the relative similarity distribution across text prompt embeddings. CPA can alleviate the lexical ambiguity of input text prompts.

**Questions:**

Since the proposed method uses both the images and the labels, is it fair to compare the proposed method with the self-supervised learning methods (e.g., SimCLR)?


**Limitations:**

As mentioned above, my main concern is fairness. The concern is mainly from the experiment settings.
In experiments, the authors compare the proposed method with the self-supervised methods that do not use any training labels (e.g., SimCLR, MoCo-V2). The proposed method uses all images and labels of training sets for transferring the representations from a multimodal model. It is not surprising that the proposed method can obtain better accuracy. On the other hand, the proposed method uses the same data as the fully supervised models and distills the knowledge from a multimodal model pre-trained on a large-scale dataset. It is expected that the proposed method outperforms the fully supervised methods.

To further showcase the proposed method, it'd be nice if the authors could demonstrate some possible capabilities:
- Using data not overlapped with the target datasets for representation transfer and showing that the proposed methods obtain superior performance on target datasets.
- Similar to the previous point. If the data mentioned above are hard to get, an easier way is to perform representation transfer on one dataset and testing on multiple target datasets
Another interesting direction might be to use less data (e.g., 1% of training data) for representation transfer and achieve similar results in the current manuscript.



**Strengths And Weaknesses:**

- Strengths:
The experiments show that the representation transfer of a large pre-trained multimodal model can provide good ImageNet linear probe accuracy while outperforming the existing self-supervised learning method (SimCLR) and closing the gap with supervised learning.

- Weaknesses:
The experiment settings seem not fair and practical.

---

> ### Author Response · Authors · 2022-08-02
> **We thank the reviewer for his/her comments and suggestions.**
>
> **Q1.** In experiments, the authors compare the proposed method with the self-supervised methods that do not use any training labels (e.g., SimCLR, MoCo-V2). The proposed method uses all images and labels of training sets for transferring the representations from a multimodal model.
>
> **A1.** We would like to highlight that the BeamCLIP proposes an **unsupervised representation transfer** method. For learning representations on ImageNet, the BeamCLIP uses unlabeled images (e.g, 1.2M image samples) and a smaller set of text prompts (e.g, 1K class names). ***Note that each image is not paired with any text prompt, but is compared with a set of text prompts.*** We would like to remind the reader of the BeamCLIP method for convenience. At first, given an unlabeled image, cross-modal similarity distribution is measured over multiple text prompt embeddings in the teacher's embedding space. Then, a student model is encouraged to mimic the cross-modal similarity distribution of the teacher model by matching these similarity distributions. To achieve effective transfer, we use anchor text embeddings by encoding text prompts.
>
> To further decouple the dependency to the class names of a dataset, we measure the transfer learning accuracy by using **random prompts**. **Figure 8 in the Appendix C.2** shows the transfer learning accuracy on CIFAR10 with regard to the number of random text prompts. As shown in **Figure 8 (a)**, the BeamCLIP is effective, while using only subset text prompts. As shown in **Figure 8 (b)**, the BeamCLIP is still effective, even though random prompts from ImageNet are used.
>
> | Method | Type | Img. Enc.   | 3 | 5 | 7 | 9 | 10 text prompts | - |
> |----------|-------|--------------|---|---|---|---|----|----|
> | CLIP (zero-shot) | Teacher | ViT-B/16 | - | -  | -  | -  | -  | 91.6 |
> | CLIP (zero-shot) | - | RN50      | - | - | -  | -   | -  | 75.6 |
> | **BeamCLIP** (unsupervised) | Student | RN50 | 83.47 | 83.25| 88.26| 91.84 | **92.10** |
>
> ***
> **Q2.** To further showcase the proposed method, it'd be nice if the authors could demonstrate some possible capabilities: Using data not overlapped with the target datasets for representation transfer and showing that the proposed methods obtain superior performance on target datasets.
>
> **A2.** To answer this comment, we have added **Table 5 "Comparison of zero-shot accuracy on ImageNet-1K"** in the revised paper. To check if the BeamCLIP also inherits the powerful zero-shot capability of CLIP, we compare zero-shot accuracy of CLIP variants on ImageNet-1K. For zero-shot measure, we use CC-3M and ImageNet-21K (12M samples) that do not overlap with ImageNet-1K (i.e., ImageNet-21K --> ImageNet-20K). ***We can see that the BeamCLIP-RN50 achieves about 57.5% zero-shot accuracy that is comparable with CLIP RN-50 (59.6%) that uses very large image-text pair data (WIT-400M).*** Even though the BeamCLIP uses only 3% data (ImageNet-21K (12M)), it can achieve the comparable zero-shot accuracy by effectively transferring the teacher's representations. ***Note that CLIP and OpenCLIP [1] provide only about 31.3% and 32.7% zero-shot accuracy with a similar amount of data (YFCC-15M).***
>
> | Method                 | Image Encoder | Training Data   | Teacher | Text Prompts | Batch Size | Epochs | ImageNet Zero-shot |
> |---------------------|--------|-----------------|---|---|---------|-----|------|
> | CLIP                     | RN50 | WIT-400M       | x | x | 32,768 | 32 | **59.6** |
> | CLIP                     | RN50 | YFCC-15M     | x| x | - | - | **31.3** |
> | OpenCLIP [1]       | RN50 | YFCC-15M     | x | x | 256 * 8 | 32 | 32.7 |
> | OpenCLIP [1]       | RN50 | CC-12M          | x | x | 256 * 8 | 32 | 36.5 |
> | **BeamCLIP** (ours) | RN50 | CC-3M | CLIP ViT-B/16 | IN-1K | 64 * 8 | 100 | 49.5 |
> | **BeamCLIP** (ours) | RN50 | IN-21K (12M) | CLIP ViT-B/16 | IN-1K | 64 * 8 | 190 | **57.5** |
>
> **References**
> [1] G. Ilharco, M. Wortsman, R. Wightman, C. Gordon, N. Carlini, R. Taori, A. Dave, V. Shankar,353
> H. Namkoong, J. Miller, H. Hajishirzi, A. Farhadi, and L. Schmidt. Openclip, jul 2021. https://github.com/mlfoundations/open_clip
>
> ***
> **Q3.** Similar to the previous point. If the data mentioned above are hard to get, an easier way to perform representation transfer on the dataset and testing on multiple target datasets. Another interesting direction might be to use less data (e.g., 1% of training data) for representation transfer and achieve similar results in the current manuscript.
>
> **A3.** Thank you for your suggestion. We think that zero-shot accuracy on ImageNet-1K explained in the above is a representative experiment, since it is large-scale. We hope that the ImageNet-1K zero-shot experiment may address your questions on the experiment setting.

---

### Official Review · Reviewer_f8ft · 2022-07-12

**Rating:** 5
**Confidence:** 3
**Soundness:** 3 good
**Presentation:** 3 good
**Contribution:** 3 good

**Summary:**

This paper presents a method to transfer multimodal representations of CLIP to smaller student networks for downstream tasks. In addition to the standard similarity loss of representations, the method also applies cross-modal similarity matching loss where the topology with respect to anchor prompts is preserved among teacher and student. It also incorporates an interesting prompt augmentation technique to deal with ambiguous or minor visual concepts. The method is tested on six benchmark datasets and showed promising performance.

Pros:
(+) The proposed method is simple yet reasonable, also seems effective.
(+) It includes an interesting prompt engineering method.
(+) The paper is generally easy to follow.

Cons:
(-) The paper seems overclaiming the improvement compared to self-supervised learning.
(-) Discussion on the limitations and negative societal impacts of the work is largely missing.


**Questions:**

Main points are described at "Strengths And Weaknesses".

Minor comments:
- According to Figure 2, in the teacher side, the gradient of CSM loss is stopped, but not for ISM loss. Is this really true? If so, what is the reason there?
- In Table 4 (ablation study), three results are missing without explanation. What do they mean? The training wasn't successful?
- The light gray font used in some tables is very hard to see.

**Limitations:**

The authors' checklist says as follows.
- (b) Did you describe the limitations of your work? [Yes] See Section 4 and Section 5.
- (c) Did you discuss any potential negative societal impacts of your work? [Yes] See Section 5.

However, I cannot find a clear discussion in the mentioned parts. I think this kind of requirements should be addressed more seriously.


**Strengths And Weaknesses:**

Overall, I find no obvious weaknesses in the paper's technical side. While each component is not surprisingly novel, the overall integration is reasonable and can be a good baseline in the research of distillating CLIP.  In particular, I find context-based prompt augmentation is an inspiring idea to successfully transfer the knowledge in CLIP to various downstream applications.

My main concern is that the paper seems to exaggerate the contribution by unfair comparison. The paper repeatedly says "outperforming the existing self-supervised learning methods (SimCLR: 65.6%)", but as shown in Table 2, with longer epochs, SwAV and BYOL-GA achieves 74.3% which is very close to the proposed method (why they are missing in Figure 1?), and even SimCLR achieves 69.1%. Why not train your method  just longer? Is it expected to improve after 200 epochs or not?
Yes, I know it is not easy to train longer with limited computational resources, but at least this point should be honestly discussed. There's no clear reason why we should stop at 200 epochs for all methods. Without such discussion, picking up a particular value of 65.6% and just saying "outperform the existing self-supervised learning methods" (L12, L67, L286) is really misguiding.

Another thing is that the limitation of the method is hardly discussed. In my opinion, context-based prompt augmentation will require some domain knowledge and engineering efforts, and yet it is not always possible to find appropriate external knowledge source for augmentation. I think this is one of clear limitations.

---

> ### Author Response · Authors · 2022-08-02
> **We thank the reviewer for his/her comments and suggestions.**
>
> **Q1.** As shown in Table 2, with longer epochs, SwAV and BYOL-GA achieves 74.3% which is very close to the proposed method. Why not train your method just longer? Is it expected to improve after 200 epochs or not?
>
> **A1.** To answer this important question, we have added **Table 2** in the revised paper. We can see that the linear probe accuracy of BeamCLIP slightly increases (from 74.8% to 75.1%), as the training epochs increase from 200 to 400. Unlike this, the accuracy of SwAV considerably increases (from 72.0% to 74.3%). With the help of CLIP, the BeamCLIP can quickly learn better representations than SSL methods. ***We think that learning better representations at short training epoch is an important contribution, since it is closely related to sustainability and energy.*** On the other hand, we also note that linear probe accuracy is highly dependent on hyperparameter search. Under a constrained computing environment, we could only conduct a very coarse-grained hyperparameter search for the linear probe algorithm (LBFGS).
>
> To make sure that the BeamCLIP learns better representations than SSL methods, we have additionally measured ImageNet-1K top-1 linear probe accuracy on ResNet-18. **Figure 1 and Table 3** in the revised paper show **linear probe accuracy on ResNet-18**. We can see that using CLIP as a teacher, the BeamCLIP can provides better linear probe accuracy than strong SSL methods including SimCLR, MoCo-v2, BYOL, and SwAV even at longer epochs (e.g., at 400 epoch). ***Also, we can see that the accuracy of the BeamCLIP increases (from 63.8% to 66.2%) as the training epoch increases.*** More importantly, the BeamCLIP provides better performance than OSS [1] that simultaneously learns and transfers representations from ResNet-50.
>
> | Method                | Teacher | Student | 100 | 200 | 400 |
> |---------------------|-----------|----------|------|------|------|
> | Supervised          | x | RN18 |           69.8           |
> | SimCLR               | x | RN18 | 47.1 | 49.9 | 51.8 |
> | MoCo-v2              | x | RN18 | 48.6 | 49.9 | 51.9 |
> | BYOL                   | x | RN18 | 44.2 | 47.5 | 46.8 |
> | SwAV                   | x | RN18 | 57.7 | 61.2 | 63.7 |
> | OSS [1]                | SSL RN50 | RN18 | 60.0 | 64.1 | 65.8 |
> | **BeamCLIP** (ours) | CLIP ViT-B/16 | RN18 | **63.8** | **64.8** | **66.2** |
>
> **References**
> [1] H. M. Choi, H. Kang, and D. Oh. Unsupervised representation transfer for small networks: I believe I can distill on-the-fly. In Advances in Neural Information Processing Systems (NeurIPS), 2021.
>
> ***
> **Q2.** Another thing is that the limitation of the method is hardly discussed.
>
> **A2.** We have added the **“Limitation” paragraph in the “Conclusion and Limitations” section**. We mentioned two issues: performance gain at long training epochs and additional effort for context-based prompt augmentation (CPA).
>
> ***
> **Q3.** According to Figure 2, in the teacher side, the gradient of CSM loss is stopped, but not for ISM loss. Is this really true?
>
> **A3.** The BeamCLIP loss consists of (1) instance similarity matching (ISM) loss and (2) cross-modal similarity matching (CSM) loss. In the teacher size, both ISM and CSM losses are stop-grad. To avoid ambiguity, we have revised **Figure 2**.
>
> ***
> **Q4.** In Table 4 (ablation study), three results are missing without explanation. What do they mean?
>
> **A4.** The dash means omitted experiments. We omitted some experiments in which the ablation setting is not helpful to understand the behavior of the BeamCLIP.

---

### Review · Ethics_Reviewer_YQDm · 2022-08-04

**Recommendation:**

The authors should use datasets that comply with the general ethical conduct: https://neurips.cc/public/EthicsGuidelines

The authors are encouraged to provide a discussion on the potential negative societal impact of the proposed method (please refer to the third paragraph of the “Ethics review” section for further details).

**Ethical Issues:**

Yes

**Ethics Review:**

In this work, the authors propose a method called BeamCLIP for transferring representations of a large teacher model into a small student model. The authors evaluate the proposed method on several publicly available datasets.

The paper contains a general ethical issue: to evaluate their method, the authors of the paper use CIFAR-10 and CIFAR-100 datasets, which are subsets of the Tiny Images dataset (https://www.cs.toronto.edu/~kriz/cifar.html). Please, note that the Tiny Images dataset was formally withdrawn by its creators (http://groups.csail.mit.edu/vision/TinyImages/).

Unfortunately, I could not find a discussion on the potential negative societal impact of the proposed method in Section 5. While training a smaller model is important resource-wise (as noted by the authors), it would be interesting to see a discussion of high societal impact applications of the method. For example, if used for facial recognition, could it be that the trained student model leads to significant performance degradation for minority demographic groups compared with the teacher model? If so, how can this problem potentially be resolved?

---

### Review · Ethics_Reviewer_JRjc · 2022-08-16

**Recommendation:**

I think this article is well written and has responded nicely to criticism. The suggestion I make above is therefore merely that (a suggestion).

**Ethics Review:**

The article does not raise any red flags per se. It would be interesting, however, to read a section that explicitly addresses the ethical implication of a study such as this. How do they relate to ethical discussions in relation to datasets such as ImageNet and the discriminatory biases in Wikipedia? And which potential positive steps does this study take in mitigating the harmful societal issues identified in other transfer learning projects?

---

### Meta-Review · Area_Chair_WGir · 2022-08-26

**Recommendation:** Accept
**Confidence:** Less certain

**Metareview:**

This paper presents a method, BeamCLIP to transfer multimodal representations of CLIP to smaller student networks for downstream tasks. It combines both similarity loss of representations and cross-modal similarity matching loss among teacher and student.
The method is tested on six benchmark datasets and showed convincing performance improvement. Reviewers' concerns on fairness and experimental comparison are properly handled in author feedback.

**Award:**

No

---

### Decision · Program_Chairs · 2022-09-14

Accept